# Effects of Ultrasound Irradiation on the Co-Pigmentation of the Added Caffeic Acid and the Coloration of the Cabernet Sauvignon Wine during Storage

**DOI:** 10.3390/foods11091206

**Published:** 2022-04-21

**Authors:** Zhendan Xue, Tingting Wang, Qing’an Zhang

**Affiliations:** Institute of Food and Physical Field Processing, School of Food Engineering and Nutrition Sciences, Shaanxi Normal University, Xi’an 710062, China; 15639292763@163.com (Z.X.); wangttedu@163.com (T.W.)

**Keywords:** ultrasound, caffeic acid, co-pigmentation, wine color, anthocyanins, storage

## Abstract

In this paper, experiments were conducted to investigate the effects of ultrasound irradiation on the co-pigmentation of caffeic acid added in wine and the coloration of wine during storage. The wine color, chroma, level of the monomeric, combined and polymerized anthocyanins and the concentrations of malvidin-3-O-glucoside and syringic acid in wines were determined by the high-performance liquid chromatography (HPLC), ultraviolet-visible spectroscopy, respectively. The results indicate that ultrasound irradiation could definitely affect the color characteristic of wine to a certain extent. Compared with the wine without addition of caffeic acid, the co-pigmentation effects of wine added with caffeic acid could be significantly promoted by ultrasound irradiation, such as the wine color, color density and the polymerized anthocyanins. Furthermore, ultrasound irradiation had a continuous effect on the co-pigmentation of caffeic acid and wine coloration with the extended storage time. In summary, ultrasound could significantly modify the color properties of wine by enhancing the co-pigmentation between caffeic acid and monomeric anthocyanins in the wine, resulting in the improvement of wine quality.

## 1. Introduction

It is well known that wine color is one of the most intuitive characteristics to define the wine quality to a large extent, which mainly depends on the contents of phenolic compounds and anthocyanins in the wines [1]. Anthocyanin is an extremely important natural water-soluble pigment [2] and mainly exists in the monomeric, combined and polymeric forms [3]. Generally, the monomeric anthocyanins are responsible for the bright red of wine color during the initial stage of fermentation [4,5], and the wine color is primarily dependent on the levels of polymerization of the anthocyanins in the later stages of wine maturation [2]. In the meantime, the stabilization of monomeric anthocyanins could be influenced by the temperature, oxygen, *p*H, light and some enzymes, etc. By combining or condensing with other phenolic compounds and polymerizing with flavan-3-ols, the co-pigmentation reactions formed more colorful and stable polymerized anthocyanins [6,7,8]. Therefore, along with the monomer anthocyanins gradually changed into the polymerized anthocyanins during the progressing of fermentation, resulting in the wine color changes from bright red to brick red, and the wine quality could also be improved [9].

In today’s highly market competition environment, wine in courtesy culture at dinner highlights the critical role that meeting the consumers’ needs plays in promoting the competitive of wine in the market [10], many researches have been carried out, such as using the oak barrel ageing [11], adding some cofactors to wines for taking co-pigmentation reactions [12], and employing some innovative physical aging technologies likely ultrasound [10,13,14], high pressure [15], microwave [16] and pulsed electric fields [17]. Although oak barrel fermentation, as a traditional and efficient method, could obtain high quality wine, it takes a lot of time, money and labor. In addition, co-pigmentation is one of the most important factors affecting the wine color, which mainly includes the anthocyanins- anthocyanins and anthocyanins-cofactor [18,19]. And cofactor is generally a colorless substance, mainly including the flavan-3-ols (catechin, epicatechin), hydroxycinnamic acids (caffeic, p-coumaric) and the hydroxycinnamoyltartaric acids (caftaric, coutaric and fertaric acids) [5]. Comparing with the oak barrel ageing, the method of adding extra cofactors is more efficient in color enhancement, which is also mainly the reaction of substances in red wine. However, on the one hand, it is established in a progressive way, not an immediate phenomenon, thus the red wine fermentation and ageing still takes a long time [20,21,22], on the other hand, whether the co-pigmentation reaction could occur and its speed are not only limited by the cofactors content, but the levels of anthocyanins [12]. Recently, how to shorten the ageing period of wine and improve the sensory characteristics of wine had attracted more attention of some researchers and manufacturers. Consequently, some innovative physical ageing technologies appeared in people’s vision, such as ultrasound technology, which has been widely used in food processing, extraction and preservation, owning to its low-cost, low-energy consumption and eco-friendly advantages [23]. Furthermore, it is also considered to be a promising technology to accelerate the ageing and obtain high quality wine within a short time [16,24]. In addition, our previous investigations have also demonstrated ultrasound as an effective technique to affect the electrical conductivity and total phenolic compounds content, promote acetaldehyde production and enhance the chromatic characteristics and stability, reduce higher alcohols, thus, modifying flavor and mouth-feel of wine to some extent [10,25,26,27].

In addition, Monagas and Bartolomé reported that the co-pigmentation in young red wine could contribute about 30–50% of the color [7], and the cofactors were the indispensable parts for the co-pigmentation. Among the cofactors, the flavan-3-ol, caffeic acids and ferulic acids were considered to be powerful cofactor in the formation of colored compounds, as a comparison, the caffeic acid was more effective than the catechin in color enhancement [21,28,29]. In addition, it has been found that pre-fermentative addition of caffeic acid increased the color and aroma compounds of dry red wines by Li et al. [30].

In a word, as above-mentioned, literatures have proved that the co-pigmentation could greatly contribute to the color changes during wine ageing, in the meantime, results also indicate that ultrasound could definitely modify the wine color. However, no information is available about the effects of ultrasound on the co-pigmentation of wine color. Since ultrasound can generate high-frequency vibration and acoustic cavitation, trigger certain chemical reactions instantaneously and accelerate the reaction rate, which might influence the complex reactions and organoleptic properties of wine [31]. Thus, the objective of this study was performed to explore the effects of ultrasound on the co-pigmentation of caffeic acid on wine, so as to promote the application of ultrasound and caffeic acid in winery to acquire high quality red wine in a short time.

## 2. Materials and Methods

### 2.1. Wine Sample and Reagent

The wine sample of 2015 vintage Cabernet Sauvignon with 12% (*v*/*v*) ethanol was supplied by Sanxian Winery (Weinan, Shaanxi Province, China) and used throughout the research process. The standards were used including malvidin-3-O-glucoside and syringic acid (Shanghai Tauto Biotech Co., Ltd., Shanghai, China), caffeic acid (National Institutes for Food and Drug Control, Beijing, China). And all the standards were dissolved in the methanol at a concentration of 1 mg/mL and stored in the dark at 4 °C before use. HPLC-grade methanol was purchased from the Fisher Scientific (Branchburg, NJ, USA). Formic acid was obtained from the Tianli Chemical Reagent Co. Ltd. (Tianjin, China). Acetaldehyde was obtained from the Kelong Chemical Reagent Factory (Chengdu City, Sichuan Province, China). Sodium pyrosulfite was obtained from the Sheng’ao Chemical Reagent Factory (Tianjin, China). All other chemicals and reagents used were of analytical grade. Ultrapure water was prepared by Millipore Milli-Q purification system (Burlington, MA, USA).

### 2.2. Ultrasound Equipment

Ultrasonic homogenizer (frequency of 20 kHz, Ningbo Science and Biotechnology Co., Ltd., Ningbo City, Zhejiang Province, China) was used for the ultrasonic treatment. The treatment time was 14 min and 28 min, respectively. The ultrasonic energy was transmitted to the sample in the beaker through the ultrasonic probe, with a total nominal output of 950 W.

### 2.3. Sample Treatment

Adding a certain amount of caffeic acid to the Cabernet Sauvignon wine, the concentration of the added caffeic acid was 1 mmol/L. The determinations were immediately carried out on the treated-wine after the treatment: each sample was treated with ultrasound for a certain time, and the temperature was kept at 16 °C with the circulating cold water. The samples were named as the following:

Sample 1 (W+Ut0): red wine without adding caffeic acid as the control sample without ultrasound;

Sample 2 (W+Ut14): red wine without adding caffeic acid treated with ultrasound for 14 min;

Sample 3 (W+Ut28): red wine without adding caffeic acid treated with ultrasound for 28 min;

Sample 4 (W+CA+Ut0): red wine with adding caffeic acid as the control sample without ultrasound;

Sample 5 (W+CA+Ut14): red wine with adding caffeic acid treated with ultrasound for 14 min;

Sample 6 (W+CA+Ut28): red wine with adding caffeic acid treated with ultrasound for 28 min.

After treatment, the samples were correspondingly stored into the six 80 mL bottles that were made of transparent plastic and covered with lid at room temperature (16–22 °C), and sealed with the parafilm to keep airtight and surrounded with aluminum foil to keep lucifugal. Consequently, on the 1st, 5th, 10th, 15th, 20th, 60th, 90th and 120th storage day after being treated [10], a certain wine samples were taken out and collected separately to measure the contents of monomeric, combined and polymerized anthocyanins, malvidin-3-Oglucoside, syringic acid and color index by HPLC, ultraviolet-visible spectroscopy, respectively. These treatments were performed in triplicate.

### 2.4. Color Analysis

According to the report of Zhang [10], samples were put into a 1.0 mm quartz cuvette and scanned by the TU-1810 UV-Vis spectrophotometer (Beijing Persee General Instrument Co., Ltd., Beijing, China) from the wave of 380 nm to 780 nm, and the color characteristics (browning index and wine color) of the samples were determined at the absorbance of 420, 520 and 620 nm, respectively. The values of color density were the sum of the absorbance at 420, 520 and 620 nm.

### 2.5. Determination of the Proportion of Monomeric, Combined and Polymerized Anthocyanins in Red Wine

20% acetaldehyde solution and sodium metabisulfite solution containing 5% sulfur dioxide were firstly prepared, then 20 μL of 20% acetaldehyde solution and 160 μL of sodium metabisulfite solution were put into 2 mL red wine samples, respectively. The absorbances of *A^acet^* and *A*^*SO*_2_^ were determined at 520 nm after balancing for 40 min, then the absorbances of *A^wine^* without acetaldehyde and sulfur dioxide was determined. Finally, the contributions of monomer, combined and polymerized anthocyanins in each red wine sample to the red wine color were calculated by the following formula [32]:$$\mathrm{The}\;\mathrm{monomer}\;\mathrm{anthocyanin}\;\mathrm{contribution}\;(\%)=\frac{A^{wine}-A^{SO_{2}}}{A^{acet}}*100$$
$$\mathrm{The}\;\mathrm{combined}\;\mathrm{anthocyanin}\;\mathrm{contribution}\;(\%)=\frac{A^{acet}-A^{wine}}{A^{acet}}*100$$
$$\mathrm{The}\;\mathrm{polymerized}\;\mathrm{anthocyanin}\;\mathrm{contribution}\;(\%)=\frac{A^{SO_{2}}}{A^{acet}}*100$$

### 2.6. Determination of Malvidin-3-O-Glucoside, Caffeic Acid and Syringic Acid in Red Wine by HPLC

The high-performance liquid chromatography system (Dalian Elite Analytical Instrument Co., Ltd., Dalian, Liaoning Province, China) was used for the determination of malvidin-3-O-glucoside, caffeic acid and syringic acid according to the literature with some modifications [10], which was equipped with a P230II binary pump, a Rheodyne injector (loop, 20 μL) and a UV230II detector (Elite, Dalian, Liaoning Province, China). The chromatogram was recorded by the EC2006 software (Elite, Dalian, Liaoning Province, China). The sample was separated using a TC-C_18_ column separation (5 μm, 4.6 mm × 250 mm, Agilent, Santa Clara, CA, USA), and the mobile phases were composed of A (H_2_O containing 1% formic acid) and B (methanol containing 1% formic acid), ultrasonically degassed for 25 min before use and filtered by the 0.45 μm membrane. The chromatographic conditions were as follows: 20 μL of injection volume, 1.0 mL/min of flow rate, and column temperature of 25 °C. The following linear gradient program was used: 0–2 min, 0–2% B; 2–5 min, 2–10% B; 5–7 min, 10–22% B; 7–12 min, 22–25% B; 12–28 min, 25–55% B; 28–35 min, 55–100% B; 35–40 min, 100–100% B; 40–45 min, 100–2% B.

### 2.7. Statistical Analysis

Experiments were repeated three times and determinations were performed in triplicate. Utilizing the SPSS (version 16.0; SPSS Inc., Chicago, IL, USA) to conduct the analysis of variance (ANOVA). Data and graph were processed using the software of Microsoft Office Excel (2017 version).

## 3. Results and Discussion

### 3.1. Effect of Ultrasound Irradiation on Visible Spectrum of Red Wine Added with Caffeic Acid

Figure 1A,B are the visible spectra of wine samples on the 1st and 120th storage day, respectively. As shown in Figure 1A, all the curve profiles of the wine samples were quite similar. Compared with the visible spectra of the samples of W+Ut0 and W+CA+Ut0, the absorbances of the later were higher than that of the former, especially in the band around 520 nm, which is in agreement with the reported results [12]. Furthermore, the spectrum of wine samples with the addition of caffeic acid could be clearly increased by the ultrasound irradiation, which might be attributed to the free radicals generated by ultrasound irradiation could induce some chemical chain reactions to accelerate polymerization or co-pigmentation, resulting in a significant increase of the absorbance, and this is similar to the trends during the normal fermentation of wine [4,13]. That is to say, the co-pigmentation reaction between caffeic acid and anthocyanins in wine could be accelerated by the ultrasonically generated free radicals. Moreover, among all the wine samples, the wine sample added with caffeic acid and subjected to ultrasonic treatment for 28 min had the highest absorbances. Compared with the wine sample stored on the 1st day, the absorption values of the wine sample stored for 120 days (Figure 1B) were higher than those of the former, and the absorptions of the ultrasonically treated samples were significantly higher than those of the untreated wines, which is consistent with the results in Figure 1A, suggesting the changing trends of the wine samples during storage are more stable. In comparison, the W+CA+Ut28 sample had the highest absorbances, indicating that the influence of the ultrasonic treatment could not only enhance the co-pigmentation of caffeic acid in short time, but also have a continuous enhancement effect on wine color.

### 3.2. Effect of Ultrasound Irradiation on the Color Parameters of Red Wine Added with Caffeic Acid

#### 3.2.1. Wine Color and Color Density

As shown in Figure 2a,b, the color and color density of wine samples treated with ultrasound are evidently higher than those of the untreated wines, and the changing trends became more obviously with the increasing of ultrasound treatment time. In addition, the wine color of the W+CA+Ut0 wine sample was significantly higher than W+Ut0 wine sample, while the color density was lower, which indicated that added caffeic acid improved wine color and color density to a certain by effecting the co-pigmentation of caffeic acid and anthocyanin. Generally, the longer the storage time, the more the derivatives of co-pigmentation reactions between the caffeic acid and monomeric anthocyanins, suggesting that the wine color increases steadily with the storage time, i.e., the co-pigmentation of caffeic acid with monomeric anthocyanins in red wine enhanced the wine color [21,33], and slightly increased the color saturation of wine, which is consistent with the results in section of Figure 1B. Furthermore, by producing a motive force to promote the changes of wine color during the irradiation process, thereby the ultrasound treatment increased the hydrophobic interaction between the caffeic acid and anthocyanins and accelerated the co-pigmentation reaction, resulting in the enhancement of wine color [34,35]. It could be seen from Figure 2b that the color density of the ultrasonically-treated wine samples were obviously increased, which might be due to the change of steric hindrance during ultrasonic treatment to prevent water molecules from attacking anthocyanins to stabilize the color of the anthocyanins. Moreover, ultrasound irradiation might also accelerate the polymerization of flavan-3-ols to produce yellow polymers, so that the red, yellow, blue and other pigments in wine were increased, and the color stability was enhanced. As a result, the wine color was continuously enhanced and the effect of ultrasonic treatment on the subsequent wine sample storage was also persistent [36].

#### 3.2.2. Browning Index

Generally, phenolic compounds are the easily oxidized substances to cause wine browning, which oxidation reactions are very complicated and mainly depend on the compositions and contents of phenols, especially the compounds with o-hydroxyphenol structure [37]. Figure 2c shows the browning indexes of each wine sample changing during storage. As shown in Figure 2c, the changing trend of the browning indexes presented an overall increase during storage, which is similar with the reported results [38]. Moreover, the browning index of the W+Ut0 and W+CA+Ut0 wine samples were closer, and the W+Ut14, W+Ut28, W+CA+Ut14 and W+CA+Ut28 wine samples were much closer, suggesting that the caffeic acid hardly affects the browning index during the storage of wine. In the meantime, browning indexes of the wines increased after ultrasonic treatment, which might suggest that the ultrasound irradiation could speed up some oxidation reactions during the wine ageing [10].

### 3.3. Effect of Ultrasonic Treatment on the Ratio of Anthocyanins with the Combined, Monomer and Polymerized State in Wine Added with Caffeic Acid

The changing trends of the combined, monomer and polymerized anthocyanins in the ultrasonically treated wine samples during storage were shown in Figure 3. As shown in Figure 3A, the changing trends of the combined anthocyanins proportion in the wine samples were all similar and presented an overall decreasing trend during the 120 storage days. Although there were some fluctuations of the proportion of combined anthocyanins in wine samples during the initial 20 storage days, then followed by a steadily decrease until the final storage day, suggesting that the co-pigmentation reaction between co-pigments and anthocyanins was more active during these storage days, and the contribution of combined anthocyanins to the wine color was becoming smaller and smaller with the increase of storage time, probably due to the reduction of monomer anthocyanin, co-pigment and other substances in wine, which made the related reaction weakened and led to wine color gradually stabilized. Compared with the proportion of the combined anthocyanins on the first storage day, the ratio of anthocyanins in adding caffeic acid wine sample was much higher than that of the non-added sample, which is similar with the report [39], suggesting that the addition of caffeic acid could increase the contribution of the anthocyanins on wine color. Furthermore, the increased proportion of the combined anthocyanins might be caused by the following two aspects: on the one hand, the decarboxylation of caffeic acid could form the ethyl or vinyl phenolic compounds during the ageing process, which was involved in the production of anthocyanins or partially molecular anthocyanin derivatives in wine [5,7]; on the other hand, phenols and anthocyanins were combined to form the π-π complexes, and then gradually transformed into more stable substances with the ageing progress [40], thus, the absorbance values and the maximum absorption wavelength were significantly increased [41], resulting in the stability of the wine color improved. In addition, the combined anthocyanins in wine ultrasonically treated for 28 min were significantly higher than those of the untreated wine on the first storage day, which might suggest that ultrasound irradiation did accelerate the combination of caffeic acid and anthocyanins, resulting in an increase of the combined anthocyanins in wine.

With respect to the monomer anthocyanins, also the mainly coloring substance of young wine [1], the proportions of monomer anthocyanins in wines were shown in Figure 3B. In all sample wines, the proportions of the monomer anthocyanins presented an overall decreasing trend during the 120 storage days, and the decrease could be attributed to the self-degradation, polymerization, oxidation or reactions with other substances [42]. In the meantime, the numerous reactions involving anthocyanins could cause the formation of new oligomeric and polymeric pigments, finally resulting in a change about the perceived wine color and enhancing the wine color quality [4,5]. Compared with the wine samples of W+Ut0 and W+CA+Ut0, the monomer anthocyanins of wine decreased after adding the caffeic acid. Therefore, it could be speculated that the adding caffeic acid to the red wine could significantly accelerated the co-pigmentation of monomeric anthocyanins and caffeic acid to form the combined anthocyanins, thereby reducing the monomer anthocyanins. Among all the wine samples, the content of monomeric anthocyanins was significantly reduced by ultrasonic treatment in the wine samples with the addition of caffeic acid, which might be due to the fact that ultrasonic treatment accelerated the co-pigmentation reactions of anthocyanins with the co-pigments or phenols to produce the structurally stable anthocyanin derivatives, i.e., accelerated the conversion of monomeric anthocyanins in a certain extent. In a word, ultrasonic treatment did promote the co-pigmentation of caffeic acid in wine samples.

Figure 3C shows the changing trend of the proportion of the polymerized anthocyanins in the wine samples during storage. For all sample wines, the proportion of the polymerized anthocyanins presented an overall increase trend during the storage, especially the proportions in wines added with caffeic acid were significantly higher than those without caffeic acid under the same ultrasonic conditions, suggesting that the addition of caffeic acid could promote the formation of polymeric anthocyanins in the wine [12]. Generally, the coloration during the ageing process is mainly caused by the anthocyanin’s polymerization. And the polymerized anthocyanins, a type of pigments with a high molecular weight, were mainly produced by the polymerization and condensation between monomeric anthocyanins and other phenolic substances, which would accumulate to form the red polymers during the natural wine ageing process, thus improving the wine color and making it more stable [43]. In addition, the longer the ultrasound time, the higher the rising of the proportion of the polymerized anthocyanins during storage. That is to say, the polymerization reaction to form polymeric anthocyanins in wine accelerated by ultrasound irradiation [40]. Overall, ultrasonic treatment did promote the co-pigmentation reactions of caffeic acid and accelerate the synthesis of polymeric anthocyanins in wine, improving the color and quality of wine, which is in agreement with the natural ageing process.

### 3.4. Effect of Ultrasonic Treatment on the Evolution of Malvidin-3-O-Glucoside and Syringic Acid of Wine with Caffeic Acid during Storage

As shown in Figure 4, the samples were separated by the HPLC with the detector at 280 nm, and identified by comparing the retention time and UV-vis spectral with the corresponding standards of the malvidin-3-O-glucoside and syringic acid, respectively. 

The contents of malvidin-3-O-glucoside and syringic acid in all wine samples were determined on the 1st, 5th, 10th, 15th, 20th, 60th, 90th and 120th storage day after ultrasound treatment. As shown in Figure 5A, the concentrations of malvidin-3-O-glucoside presented an overall decreasing trend during the 120 storage days, which is very similar with that of the monomer anthocyanins in Figure 3B. Specifically, compared to the W+Ut0 wine sample, the content of the malvidin-3-O-glucoside decreased quickly in the W+CA+Ut0 wine samples, suggesting adding caffeic acid could accelerate the co-pigmentation of malvidin-3-O-glucouside, which is in agreement with the reported results [5,7]. Furthermore, the contents of the malvidin-3-O-glucoside in ultrasound-treated wine were lower than that of the untreated wine, meaning that ultrasound irradiation might accelerate the oxidation, self-degradation and co-pigmentation reactions between malvidin-3-O-glucoside and other phenolic compounds [5,42,43], resulting in the improvement of wine quality and commercial value.

With respect to the phenolic compounds, the concentrations of syringic acid in wine samples were determined and presented in Figure 5B. The changing trend of the syringic acid contents was very similar in all wine samples and presented an overall increasing during the storage, which is also consistent with the reported results [10]. To be specific, the rising trend of the syringic acid in the ultrasonically treated wine is more obvious than that of the untreated during the whole storage time, suggesting that the ultrasound did accelerate the production of syringic acid, thus comparing with the natural ageing process of wine, the degradation of malvidin-3-O-glucoside was higher than that of the untreated samples to a certain extent.

## 4. Conclusions

In a word, the color parameters of the Cabernet Sauvignon wine with adding caffeic acid could be definitively modified by ultrasound irradiation during storage. To be specific, ultrasound irradiation significantly enhanced the color and color density of wine samples added with caffeic acid, meanwhile, markedly promoted the formation of combined and polymerized anthocyanins, and monomer anthocyanins decreased correspondingly during the whole storage time, suggesting that ultrasound treatment might accelerate the co-pigmentation between the caffeic acid and the monomer anthocyanins. In summary, all the results suggest that the ultrasound irradiation could definitely promote the co-pigmentation between the caffeic acid and the anthocyanins, modifying the color quality and stability of wine. Furthermore, the color indexes and anthocyanins coloration also improved by adding caffeic acid to Cabernet Sauvignon wine. However, the mechanism of ultrasound promoting the co-pigmentation of caffeic acid and the effects of adding extra caffeic acid on the taste and aroma components of wine remain unknown, which should be further investigated in future.

## Figures and Tables

**Figure 1 foods-11-01206-f001:**
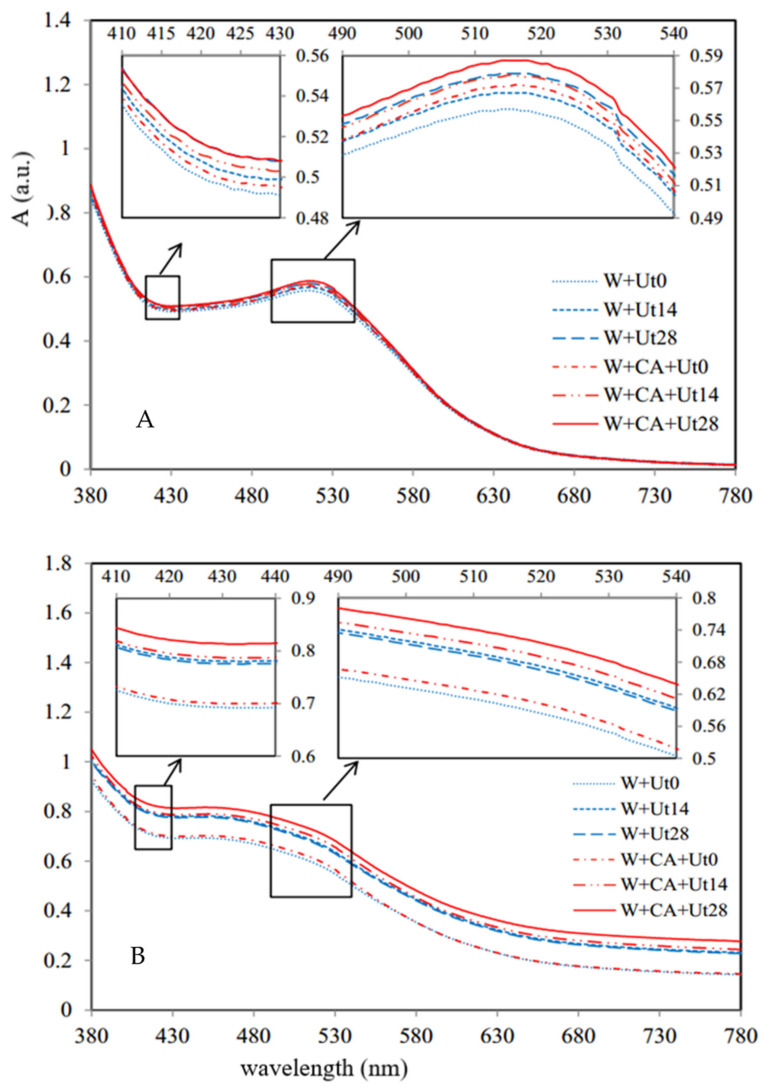
Visible spectra of red wine samples of W+Ut0, W+Ut14, W+Ut28, W+CA+Ut0, W+CA+Ut14, W+CA+Ut28 on the 1st (**A**) and 120th (**B**) storage day.

**Figure 2 foods-11-01206-f002:**
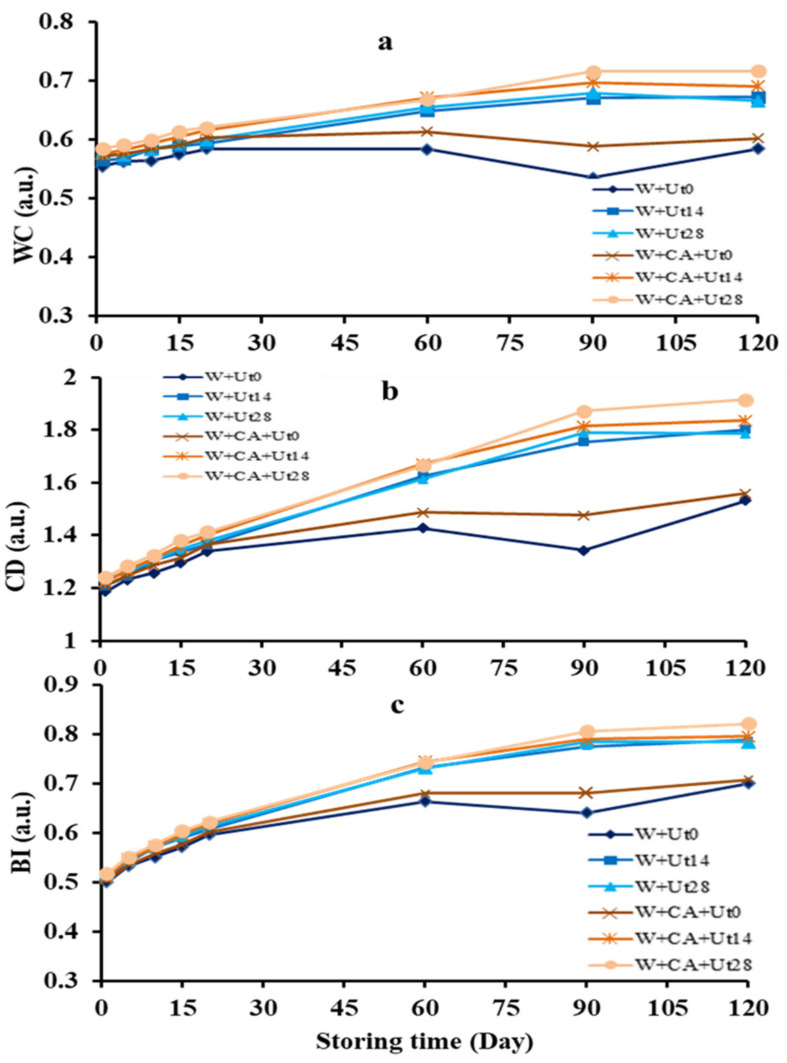
Evolutions of the WC ((**a**); wine color), CD ((**b**); color density) and BI ((**c**); browning index) of red wine samples of W+Ut0, W+Ut14, W+Ut28, W+CA+Ut0, W+CA+Ut14, W+CA+Ut28 during 120 days storage.

**Figure 3 foods-11-01206-f003:**
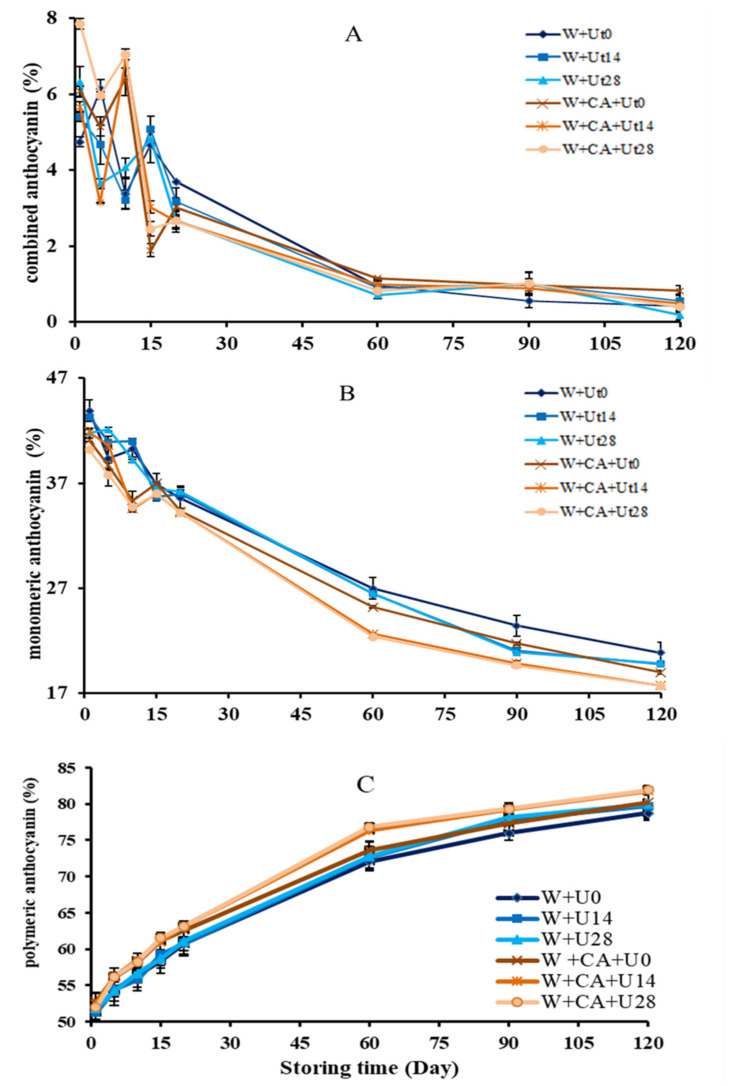
Evolutions of combined (**A**), monomeric (**B**) and polymeric (**C**) anthocyanins in wine samples during storage.

**Figure 4 foods-11-01206-f004:**
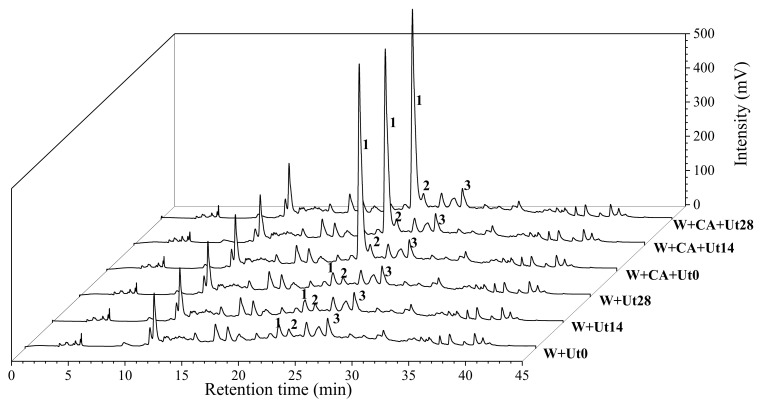
Chromatograms of wine samples of W+Ut0, W+Ut14, W+Ut28, W+CA+Ut0, W+CA+Ut14, W+CA+Ut28 obtained by HPLC at 280 nm. Peak 1: Caffeic acid (22.36 min); Peak 2: Syringic acid (23.23 min); Peak 3: Malvidin-3-O-glucoside (26.66 min).

**Figure 5 foods-11-01206-f005:**
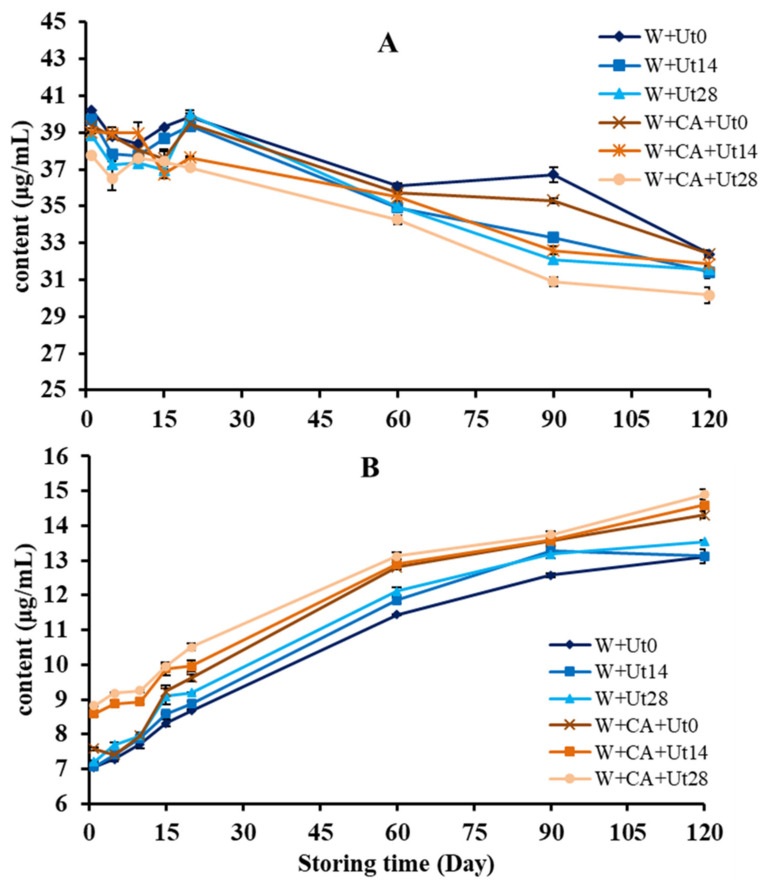
Evolutions of malvidin-3-O-glucoside (**A**) and syringic acid (**B**) in wine samples during storage.

## Data Availability

The datasets generated for this study are available on request to the corresponding author.

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
