# Peer review of "Effects of Ultrasound Irradiation on the Co-Pigmentation of the Added Caffeic Acid and the Coloration of the Cabernet Sauvignon Wine during Storage"

_foods, 2022, doi:10.3390/foods11091206_

Round 1

Reviewer 1 Report

The topic is very interesting and of great relevance. In the meantime, a review of the writing of the manuscript, and  English spelling, is necessary.

The manuscript presents the following problems:
1. The introduction must describe the objective of this manuscript;
2. Material and methods of color analysis: How is wine color assessed? How do you calculate Browning index?
3. Material and methods in Statistical analysis: What statistical analysis was used? The results didn’t include statistical analysis.
4. Results and discussion: Figure 1 should be self-explanatory.
5. Results and discussion: Figure 2  what are the values of Y axis? Better identify this figure with other letters. How do you calculate the wine color and browning index? The discussion is not clear because there was a similar curve trend for wine color and  browning index.
6. Results and discussion: Figure 4 does not clear.  What sampling  time was used?  What are the results you want to show with this figure?
7. Results and discussion: Figure 5 is self-explanatory and should be used as a model for the other figures.
8. Conclusion: Must be rewritten. It is too long. Besides, that must analyse the results for writing.

Author Response

    We greatly thank for the reviewer’s very careful work and helpful suggestions. We have been carefully revised this manuscript and have made correction according to the reviewer’s specific suggestions:

  1. The introduction must describe the objective of this manuscript.

Thanks for the reviewer’s specific and professional comments. We have carefully modified the manuscript according to the reviewer’s suggestion.

  1. Material and methods of color analysis: How is wine color assessed? How do you calculate Browning index?

Thanks a lot for the reviewer's professional comments. The wine color (WC) and browning index (BI) were the absorbance value of the sample at 520 nm and 420 nm, respectively. We have been supplemented a reference about test method in the revised manuscript.

  1. Material and methods in Statistical analysis: What statistical analysis was used? The results didn’t include statistical analysis.

Thanks a lot for the reviewer’s specific work and comments. The statistical analysis utilized the SPSS (16.0; SPSS Inc., Chicago, IL, USA) to conduct the analysis of variance (ANOVA), and we have been modified in the revised manuscript.

  1. Results and discussion: Figure 1 should be self-explanatory.

We greatly appreciate the reviewer’s professional work. The Figure 1 has been modified in the revised manuscript.

  1. Results and discussion: Figure 2 what are the values of Y axis? Better identify this figure with other letters. How do you calculate the wine color and browning index? The discussion is not clear because there was a similar curve trend for wine color and browning index.

Thanks a lot for the reviewer’s specific work and comments. I’m sorry, the figure had a question, and we have been modified in the revised manuscript.

  1. Results and discussion: Figure 4 does not clear. What sampling time was used? What are the results you want to show with this figure?

Thanks a lot for the reviewer's professional comments. The figure 4 had been modified, the retention time of caffeic acid, syringic acid and malvidin-3-o-glucoside were 22.36 min, 23.23 min and 26.66 min respectively and we had been changed the figure title in the revised manuscript. Finally, according to the condition of HPLC found that separation effect of the three substances (caffeic, syringic acid and malnidin-3-o-glucoside) was good, thus the figure 5 mainly calculated the contents of syringic acid and malvidin-3-o-glucisde by curves of corresponding standard substances (shown in Tab. 1).

Tab. 1 Curves of standards

Correspond standard substance

curves

R2

Malvidin-3-o-glucoside (280 nm)

y = 0.0386x - 0.4114

0.9990

Caffeic acid

y = 0.0170x - 0.4834

0.9992

Syringic acid

y = 0.0167x - 0.0997

0.9993

  1. Results and discussion: Figure 5 is self-explanatory and should be used as a model for the other figures.

We greatly appreciate the reviewer’s professional work. We have been modified the conclusion in the revised manuscript.

  1. Conclusion: Must be rewritten. It is too long. Besides, that must analyse the results for writing.

Thanks a lot for the reviewer's professional comments. We have been modified the conclusion in the revised manuscript.

Anyway, we greatly appreciate the reviewer’s professional work.

Reviewer 2 Report

Please improve the English of manuscript since it makes it difficult to read it.

Thorough out the manuscript authors used as synonyms the words: conjugated, combined and co-pigmentation and they are not synonyms. Please check and unify.

There is no standard deviation informed in the different figures (Figure 1 to Figure 3) and no statistical analysis was performed in any of the analysis. This is lack of scientific rigor. It is not appropriate to draw conclusions when there is no formal statistical analysis of results.  

line 28: combination with?? complete the sentence

line 48: intramolecular pigmentation? what do you mean?

line 49 – 52: is just an extension of the previous sentence.

line 52 replace Hydroxycinnamic acids by hydroxycinnamic acids

line 63: references?

line 64-66: references?

line 75 – 78: “Among the cofactors, the flavan-3-ol, such as catechin or 75 epicatechin, is considered to be a powerful cofactor in the formation of colored com- 76 pounds[27]”. This sentence is the opposite of the one that follows. Some explanation? justification?

As a comparison, at the same concentration, the hydroxycinnamic acids or 77 cinnamic acids (such as caffeic acid, erucic acid or ferulic acid) are more effective than the 78 flavonols in color enhancement[21, 28]

line 91: bound to? in other parts of introduction you mention combined…please unify.

material and methods

2.2. Ultrasound equipment: Have you controled the temperature during the ultrasonic treatment? This is important since it is known that ultrasound increase temperatura and that temperatura affects the copigmentation of antochyanins.

line 116-118: please revise the sentence. what do you mean with balance? store in darkness for how long?

why do you decided to measure the leves of malvidin-3-Oglucoside and syringic acid only? why not measuring other anthocyanins?

what is the difference between the color index and de color analysis? references?

line 148: Determination of the proportion of monomeric, combined and polymerized anthocyanins red wine

2.6. Determination of malvidin-3-O-glucoside, caffeic acid and syringic acid in red wine by 160 HPLC

Please explain how did authors identify and quantify the diferent compounds?

Authors should do some statistical analysis along dthe manuscriot to certainly state if the observed differences are significant.

Results: In this part authors should make some statistical analysis to establish if differences observed (for example in fig 1, fig 2 ) are significant.

Effect of ultrasound irradiation on visible spectrum of red wine added with caffeic acid

In this part authort sstate that ultrasound acelerates the copigmentation wth caffeic acid, however it seems taht ultrasound by it self also generates some types of reactions. Aplication of ultrasound for 28 min  by its self makes similar changes to the ultrasound for 14 min + the addittion of caffeic acid. Some explanantion to this?

Furthermore ultrasound had effects along the time by itself. Differences between samples seem to be lower at day 120 than day 0. Please explain this phenomenon.

Browning index: this index is not explain in material and method section.

How do authors explain results for browning index for W+U0 at 90 days?

The release of syringic acid generally occurs as a consequence of basic hydrolysis. What reaction do authors think is taking place between the anthocyanins and the caffeic so that the release of syringic acid acid occurs?

Author Response

Reviewer #2

We greatly thank for the reviewer’s very careful work and helpful suggestions. We have been carefully revised this manuscript and have made correction according to the reviewer’s specific suggestions:

  1. Thorough out the manuscript authors used as synonyms the words: conjugated, combined and co-pigmentation and they are not synonyms. Please check and unify.

Thanks a lot for the reviewer’s specific work and comments. We have modified in the revised manuscript.

  1. There is no standard deviation informed in the different figures (Figure 1 to Figure 3) and no statistical analysis was performed in any of the analysis. This is lack of scientific rigor. It is not appropriate to draw conclusions when there is no formal statistical analysis of results.

Thanks a lot for the reviewer’s specific work and comments. Fig. 1 was a continuous spectrum scanning diagram, which has been calibrated with water as a control, and the three scanning diagrams were repeated, so there is no statistical analysis. Because the instrument could be accurate to four decimal places, with higher precision and small error, thus, the error line is small or no in the figure 2. In addition, we have been changed the figure 3.

  1. line 28: combination with? complete the sentence.

Thanks a lot for the reviewer’s specific work and comments, and we have been modified in the revised manuscript.

  1. line 48: intramolecular pigmentation? what do you mean?

Thanks a lot for the reviewer’s specific work and comments. We have been changed in the revised manuscript.

  1. line 49 – 52: is just an extension of the previous sentence.

Thanks a lot for the reviewer’s specific work and comments. We have been changed in the revised manuscript.

  1. line 52 replace Hydroxycinnamic acids by hydroxycinnamic acids.

Thanks a lot for the reviewer’s specific work and comments. The “Hydroxycinnamic acids” in line 52 had been changed by “hydroxycinnamic acids” in line 49 of foods-1656823 in the revised manuscript.

  1. line 63: references?

Thanks a lot for the reviewer’s specific work and comments. We have been supplemented a reference (Sono-physical and sono-chemical effects of ultrasound: Primary applications in extraction and freezing operations and influence on food components. Ultrasonics Sonochemistry, 2020, 60: 10472) in the revised manuscript.

  1. line 64-66: references?

Thanks a lot for the reviewer’s specific work and comments, and we have been modified in the revised manuscript.

  1. line 75 – 78: “Among the cofactors, the flavan-3-ol, such as catechin or epicatechin, is considered to be a powerful cofactor in the formation of colored compounds [27]”. This sentence is the opposite of the one that follows. Some explanation? justification?

Thanks a lot for the reviewer’s specific work and comments. Normally, flavan-3-ols, such as (+)-catechin or (-)-epicatechin are recognized as powerful cofactors, which can form colored complexes most easily and intensely. However, our previous investigation has demonstrated that the co-pigmentation of caffeic acid was better than that of catechin[1]. And we have been modified in the revised manuscript.

  1. As a comparison, at the same concentration, the hydroxycinnamic acids or 77 cinnamic acids (such as caffeic acid, erucic acid or ferulic acid) are more effective than the 78 flavonols in color enhancement [21, 28].

Thanks a lot for the reviewer’s specific work and comments, we have been modified in the revised manuscript.

  1. line 91: bound to? in other parts of introduction you mention combined…please unify.

Thanks a lot for the reviewer’s specific work and comments, we have been carefully modified in the revised manuscript.

  1. “materials and methods” in Lines 95 of foods-1656823 have been changed by “material and methods” in the revised manuscript.
  2. 2.2. Ultrasound equipment: Have you controled the temperature during the ultrasonic treatment? This is important since it is known that ultrasound increase temperatura and that temperatura affects the copigmentation of antochyanins.

Thanks a lot for the reviewer's professional comments. In the process of ultrasonic treatment, a thermometer had been used to measure the temperature. In each experiment, the water temperature was controlled by circulating the cooling water during ultrasound treatment.

  1. line 116-118: please revise the sentence. what do you mean with balance? store in darkness for how long?

Thanks a lot for the reviewer's professional comments. I’m sorry about this. What we want to express was that the standard of caffeic acid should be kept away from light at 4 oC before use. We have been modified in the revised manuscript.

  1. why do you decided to measure the levels of malvidin-3-Oglucoside and syringic acid only? why not measuring other anthocyanins?

Thanks a lot for the reviewer's professional comments. Due to the malvidin-3-o-glucoside was the main monomer anthocyanin in wine and syringic acid was the degradation product of malvidin-3-o-glucoside during ageing, therefore, we measured the levels of malvidin-3-o-glucoside.

  1. what is the difference between the color index and de color analysis? references?

Thanks a lot for the reviewer's professional comments. Color index refers to the index effecting color, such as wine color, color density and browning index. Color analysis refers to the analysis based on the color index to judge the impact of adding caffeic acid to red wine or ultrasound, as described in the report[2].

  1. line 148: Determination of the proportion of monomeric, combined and polymerized anthocyanins red wine.

Thanks a lot for the reviewer’s specific work and comments, we have been modified in the revised manuscript.

  1. 2.6. Determination of malvidin-3-O-glucoside, caffeic acid and syringic acid in red wine by HPLC.

Thanks a lot for the reviewer’s specific work and comments, we have been modified in the revised manuscript.

  1. Please explain how did authors identify and quantify the different compounds?

Thanks a lot for the reviewer's professional comments. First of all, we separated the substances with HPLC, then prepared the standard curve of the corresponding standard compounds respectively, finally, identified and quantified these substances according to retention time and standard curves of the standards.

  1. Authors should do some statistical analysis along the manuscript to certainly state if the observed differences are significant.

Thanks a lot for the reviewer’s specific work and comments. The answer was the same as question 2 and we have been modified in the revised manuscript.

  1. Results: In this part authors should make some statistical analysis to establish if differences observed (for example in fig 1, fig 2) are significant.

Thanks a lot for the reviewer’s specific work and comments. The answer was the same as question 2 and we have been modified in the revised manuscript.

  1. Effect of ultrasound irradiation on visible spectrum of red wine added with caffeic acid

Thanks a lot for the reviewer’s specific work and comments, we have been modified in the revised manuscript.

  1. In this part author state that ultrasound accelerates the co-pigmentation with caffeic acid, however it seems that ultrasound by itself also generates some types of reactions. Application of ultrasound for 28 min by its self makes similar changes to the ultrasound for 14 min + the addition of caffeic acid. Some explanation to this?

Thanks a lot for the reviewer's professional comments. First of all, compared to untreated wine sample, ultrasound or adding caffeic acid to wine sample could accelerate the co-pigmentation, suggesting both the ultrasound and adding extra caffeic acid had some influence on wine color. In addition, the change trends of W+Ut28 and W+CA+Ut14 wine samples were similar, indicating that adding extra caffeic acid to wine sample could significantly accelerate co-pigmentation, meanwhile, ultrasound also has a powerful influence on the co-pigmentation of caffeic acid.

  1. Furthermore ultrasound had effects along the time by itself. Differences between samples seem to be lower at day 120 than day 0. Please explain this phenomenon.

Thanks a lot for the reviewer’s specific work and comments. As shown in figure 1, it could be found that the absorption values of the wine sample stored for 120 days (Fig. 1B) were higher compared to the wine sample stored on the 1st day, and the absorptions of the ultrasonically treated samples were significantly higher than those of the untreated wines, which is consistent with the results in Fig. 1A, suggesting the changing trends of the wine samples during storage are more stable. In comparison, the W+CA+Ut28 sample had the highest absorbances, indicating that the influence of the ultrasonic treatment could not only enhance the co-pigmentation of caffeic acid in short time, but also have a continuous enhancement effect on wine color.

  1. Browning index: this index is not explain in material and method section.

Thanks a lot for the reviewer’s specific work and comments. The wine color (WC) and browning index (BI) were the absorbance value of the sample at 520 nm and 420 nm, respectively. We have been modified in the revised manuscript.

  1. How do authors explain results for browning index for W+U0 at 90 days?

First of all, I’m sorry, there was something wrong with the figure uploaded before and we have been modified in the revised manuscript. Meanwhile, thanks a lot for the reviewer's professional comments. The decrease of the browning index of W+U0 wine sample at 90 th day might attribute to the colored pigment from oxidation reaction formation was destroyed, thus, we would continue to explore the specific impact in the further research.

  1. The release of syringic acid generally occurs as a consequence of basic hydrolysis. What reaction do authors think is taking place between the anthocyanins and the caffeic so that the release of syringic acid occurs?

Thanks a lot for the reviewer's professional comments. Syringic acid is the degradation product of malvidin-3-o-glucouside in the ageing process, the accumulation increased gradually with the increase of time. In addition, there was a polymerization reaction between caffeic acid and malvidin-3-o-glucouside, and promoting polymerization might affect the accumulation of syringic acid, that is, the degradation of anthocyanin. It could be seen from the figure 5 that ultrasound or adding caffeic acid to the wine sample significantly decreased the concentration of anthocyanin, but the addition of caffeic acid markedly promoted the formation of syringic acid, that is to promote the degradation of anthocyanins. At the same time, compared with adding caffeic acid to the wine sample, ultrasound was mainly to promote co-pigmentation of malvid-3-o-glucouside.

Again, special thanks to the reviewer for the very careful and constructive suggestions.

[1] Z.-D. Xue, Q.-A. Zhang, T.-T. Wang, Co-pigmentation of caffeic acid and catechin on wine color and the effect of ultrasound in model wine solutions, Journal of AOAC INTERNATIONAL. 104 (2021) 1703–1709.

[2] Q.-A. Zhang, T.-T. Wang, Effect of ultrasound irradiation on the evolution of color properties and major phenolic compounds in wine during storage, Food Chemistry. 234 (2017) 372-380.

Round 2

Reviewer 2 Report

Authors have adressed all the items asked.